



# Variability and extremes: Statistical validation of the AWI-ESM

Justus Contzen[1,2], Thorsten Dickhaus[3], and Gerrit Lohmann[1,2]

[1]Alfred Wegener Institute Helmholtz Centre for Polar and Marine Research, Bremerhaven, Germany
[2]Department of Environmental Physics, University of Bremen, Bremen, Germany
[3]Institute for Statistics, University of Bremen, Bremen, Germany

**Correspondence:** Justus Contzen (justus.contzen@awi.de)

**Abstract.** Coupled general circulation models are of paramount importance to assess quantitatively the magnitude of future climate change. Usual methods for validating climate models include the evaluation of mean values and covariances, but less attention is directed to the evaluation of extremal behaviour. This is a problem because many severe consequences of climate changes are due to climate extremes. We present a method for model validation in terms of extreme values based on classical

extreme value theory. We further discuss a clustering algorithm to detect spacial dependencies and tendencies for concurrent extremes. To illustrate these methods, we analyse precipitation extremes of the AWI-ESM global climate model compared to the reanalysis data set CRU TS4.04. The methods presented here can also be used for the comparison of model ensembles, and there may be further applications in palaeoclimatology.

## 1 Introduction

Coupled general circulation models are frequently utilised to assess quantitatively the magnitude of future climate change. Validating these models by simulating different climate states is essential for understanding the sensitivity of the climate system to both natural and anthropogenic forcing. Usual methods for validating climate models include the evaluation of mean values and covariances and the comparison of empirical cumulative distribution functions. These analyses can also be conducted over seasonal and annual averages (climatologies) or along latitudinal/longitudinal transects (Tapiador et al., 2012).

The comparison of climate indices is also common in model validation (Sillmann et al., 2013; Zhang et al., 2011). While climate models are able to reproduce many climate phenomena across the globe, their reliability regarding extremes requires additional evaluation. Changes in the intensity and frequency of extremes have drawn much attention during recent decades (IPCC, 2012; Rahmstorf and Coumou, 2011; Horton et al., 2016), mainly due to their large impacts on natural environment, economy and human health (Ciais et al., 2005; Kovats and Kristie, 2006). For instance, the summer heat wave over Central

Europe in 2003 resulted in extensive forest fires, crop yield reductions and fatalities (de Bono et al., 2004; Vandentorren et al., 2004). During the 20th century, the frequency of high-temperature extremes has increased in Europe (Dong et al., 2017), even after the apparent levelling off of global mean temperatures after 2000 (Trenberth and Fasullo, 2013), and for precipitation extremes, a similar development has been observed (Fischer and Knutti, 2016). Due to the inherent nature of extreme events, their evolution differs from that of the mean and the variance (Schär et al., 2004; IPCC, 2012) and also highly depends on the





strength of the events themselves (Myhre et al., 2019).

In particular, the concurrent occurrence of climate extremes at different locations may have especially large impacts on agriculture (Toreti et al., 2019), human societies and economies (Jongman et al., 2014) and on the climate system itself (Zscheischler et al., 2014). Large-scale climate extremes can furthermore cause serious problems for insurance and reinsurance companies (Mills, 2005). For these reasons, an increasing amount of research is being conducted on multivariate analysis of extremes

with focus on their concurrent appearance (Shaby and Reich, 2012; Dombry et al., 2018; Kornhuber et al., 2020; Ionita et al., 2021a) and new tools have been created for the analysis of extremes in climate models (Weigel et al., 2021).

The analysis of extreme events is often complicated by the fact that extreme events are typically rare, and that it is therefore

difficult to build informative statistics based solely on the extreme events themselves. As a remedy, it is common to apply the block-maxima approach, i.e. to group data into blocks of a sufficiently large size and investigate the block-wise maxima. Under suitable conditions, these block-wise maxima can be described with the generalised extreme value (GEV) distribution. In this work, we will evaluate the performance of the fully coupled AWI Earth System Model AWI-ESM1.1LR in terms of its accuracy regarding variability and extremes of precipitation, putting special focus on spatially concurrent precipitation ex-

tremes. Our main questions are whether the model is able to correctly reproduce extreme events in different regions and if spatial dependencies and concurrent extremal events are correctly identified. We compare model data from a historical run of the AWI-ESM to the global precipitation reanalysis data set CRU TS4.04. We start with investigating variability and extremes locally using empirical statistics and by fitting a GEV distribution to annual precipitation maxima. Then, following an approach by Bernard et al. (2013), we use a clustering algorithm to group spatio-temporal climate data into different spatial regions based

on their similarity in terms of extremal behaviour and the concurrency of their extremes. This clustering is based on the theory of max-stable copulae, which has been used extensively to investigate spatial dependence of extreme precipitation events, for example in Bargaoui and Bárdossy (2015); Zhang et al. (2013); Qian et al. (2018). In those papers, an analysis of bivariate variables is performed. In our work, we first construct for each pair of locations a measure for their similary in terms of extremes. This measure is then used as a basis for the clustering algorithm to group the data into spatial regions of comparable extremal

behaviour. The resulting clusters for model and observational data are compared and used to analyse the ability of the climate model to correctly reproduce spatial dependencies of precipitation extremes.

Applying clustering algorithms to climate data is not a new approach. Among others, it has been used to define climate zones in the United States (Fovell and Fovell, 1993) and globally (Zscheischler et al., 2012), and to find regions with similar

trends in their climatic change over Europe (Carvalho et al., 2016). Those analyses focus on mean values and on their temporal differences, respectively, while we apply clustering specifically to uncover connections regarding climate extremes.

The article is structured as follows: After introducing the data sets in Sect. 2, we present the methods used in Sect. 3. The results from their application to the data are presented in Sect. 4. A section on conclusions and discussions finalises the article.





## 2 Data


The observational data are reanalysed monthly precipitation data in $\mathrm{mm}$ over land (excluding Antarctica) from the CRU TS4.04 data set (Harris et al., 2020; University of East Anglia Climatic Research Unit et al., 2020) with data ranging from 1901 to 2019. We restrict the time frame to the years 1930 to 2014 in order to have a sufficiently large area with non-missing data and to be consistent with the climate model data. The grid size is $0.5° \times 0.5°$, the data have been obtained by interpolating

observations from more than 4.000 weather stations using angular distance weighting.

At some locations and time points, no data from nearby weather stations was available to use for interpolation. In these cases, the creators of the CRU TS4.04 data set used a value from a climatology instead. These climatology values are uninformative in terms of extremes and too many of them would distort the analyses, therefore all grid points with more than $5\%$ climatology values and additionally all grid points with at least 12 consecutive months of climatology values are excluded from our analy-

sis. This results in the exclusions of larger regions in northern and central Africa, in Indonesia, in central Asia and in the polar regions. In the figures below, those regions are coloured in grey.

The climate model used is the coupled model AWI-ESM1.1LR. It is based on the AWI Earth System Model (AWI-ESM1), which consists of the AWI Climate Model (Sidorenko et al., 2015; Rackow et al., 2018), but with interactive vegetation. The

model comprises the atmosphere model ECHAM6 (Stevens et al., 2013), which is run with the T63L47 setup, that is, a horizontal resolution of $1.85° \times 1.85°$. The ocean-sea ice model FESOM1.4 (Wang et al., 2014) employs an unstructured grid, allowing for varying resolutions from $20\mathrm{km}$ around Greenland and in the North Atlantic to around $150\mathrm{km}$ in the open ocean (CORE2 mesh). The land surface processes are computed by the land surface model JSBACH2.11 (Reick et al., 2013). The model considers the surface runoff toward the coasts, deploying a hydrological discharge model that also includes freshwater

fluxes by snowmelt (Hagemann and Dümenil, 1997).

AWI-ESM1 has been extensively used and described in the context of palaeoclimate changes as well as of changes of the recent and future climate (Shi et al., 2020; Lohmann et al., 2020; Ackermann et al., 2020; Niu et al., 2021). The historical run is documented in Danek et al. (2020) and has been directly used in Ackermann et al. (2020) and Keeble et al. (2021). The

model takes furthermore part in CMIP6/PMIP4 activities (Brierley et al., 2020; Brown et al., 2020; Otto-Bliesner et al., 2021; Kageyama et al., 2021a, b).

In our analysis, the time frame is restricted to the years 1930 to 2014, as in the observational data. We investigate monthly precipitation (sum of convective precipitation and large-scale precipitation) in $\mathrm{mm/month}$. We use bilinear interpolation to

scale the data to the $1° \times 1°$ grid of the reanalysis data set and take into account only those interpolated grid points that correspond to locations with given observed data, excluding the oceans and the regions with incomplete data mentioned above.





## 3  Methods

### 3.1  Univariate Analysis

In this subsection, the time series of all data points are investigated separately, and all operations and analyses described are therefore conducted for each grid point. Since the focus of this work is not on evaluating the effects of long-time trends, we apply a seasonal-trend decomposition using Loess (Cleveland et al., 1990) on the data and subtract the deviance of the trend from its mean value from it, resulting in data that can be assumed time-stationary. Then, as a first comparison between the data sets, we investigate differences in the empirical mean and empirical standard deviation of the annually maximised precipitation data.


The theoretical foundation for the application of the GEV distribution is as follows: For a random variable $X$ with an unknown probability distribution, we look at the distribution of the maximum of i.i.d. copies $X_1, \ldots, X_n$ of it: $Y^{(n)} :=$ $\max_{i=1,\ldots,n}(X_i)$. We assume that for suitable normalising sequences $a_n > 0$ and $b_n$, these blockwise maxima converge in distribution if the block size $n$ tends to infinity:

$$\frac{Y^{(n)} - b_n}{a_n} \xrightarrow{\mathcal{D}} H. \tag{1}$$

In this case, as shown by Fréchet (1927), Fisher and Tippett (1928) and Gnedenko (1943), the distribution of $Y^{(n)}$ can be approximated by a GEV distribution for a large (fixed) value of $n$. This distribution depends on the three parameters location ($\mu$), scale ($\sigma > 0$) and shape ($\gamma$) and its cumulative distribution function is given by

$$F_{\mu,\sigma,\gamma}(x) = \begin{cases} \exp(-\exp(-\frac{x-\mu}{\sigma})) & \gamma = 0 \\ \exp(-\max(0, 1 + \gamma \frac{x-\mu}{\sigma})^{-\frac{1}{\gamma}}) & \gamma \neq 0 \end{cases} \tag{2}$$

The GEV distribution has widely been used as a model for blockwise maximised data, especially for the yearly maximum of daily average precipitation (Coles et al., 2003; Onwuegbuche et al., 2019; Villarini et al., 2011, for example). Following this approach, we group our monthly precipitation data from observations and climate model into one-year block maxima and fit a GEV distribution to each grid point. When selecting a block size, a bias-variance tradeoff has to be taken into account: For a low block size, the resultung parameter estimates tend to be biased because the convergence to the GEV distribution holds only

asymptotically. A high block size, on the other hand, will lead to a limited amount of block-wise maxima that can be analysed and therefore to a higher variance in the estimates (see McNeil et al. (2015), Chapter 7). In our case, we have a relatively small block size of 12 (months per year) and a number of block-wise maxima of 90 (years of investigation).

To estimate the distribution parameters, we use the method of Probability-Weighted Moments developed by Hosking (1985)

as implemented in the R package "EnvStats" of Millard (2013). As shown by Hosking et al. (1985), this method yields estimators with a relatively low variance and bias compared to the maximum likelihood approach, especially for small and medium-size samples. We test the goodness of fit using a one-sided Kolmogorov-Smirnov-test at significane level 5%.





For each parameter, we also compute $95\%$ confidence intervals using the parametric bootstrap method with 2500 resamples.

## 3.2 Comparison of spatial distributions

To compare the spatial distributions of climate extremes, we introduce a hierarchical clustering algorithm (using average linking) to determine regions with similar extremal behaviour. This approach is similar to the idea proposed in Bernard et al. (2013). For hierarchical clustering, an appropriate dissimilarity function $D : A \times A \to \mathbb{R}$ with $A$ the total set of grid points is required. This function must fulfil $D(x,x) = 0$ and $D(x,y) = D(y,x) \geq 0$ for all $x,y \in A$.

One possible dissimilarity measure is based on the extremal coefficient $\theta_{x,y}$, a measure of the strength of the dependency of GEV distributed attaining a low value if the extremes in the distributions at $x$ and $y$ tend to be concurrent. The extremal coefficient is based on the theory of max-stable copulae: Assume that the two real-valued random variables $(X,Y)$ have a copula function $C : [0,1] \times [0,1] \to [0,1]$, that is, their joint distribution function can be written in terms of the copula and the marginal distribution functions as $F_{X,Y}(x,y) = C(F_X(x), F_Y(y))$ for all $x,y \in \mathbb{R}$. Then, if $(X,Y)$ is the limit of block-wise maxima of a sequence of i.i.d. two-dimensional variables when the blocksize goes to infinity (a similar condition as in Sect. 3.1, extended to two-dimensional random variables), it follows immediately that $X$ and $Y$ are GEV distributed. As shown for example in McNeil et al. (2015), Theorem 7.44 and 7.45, the copula must also fulfil $C(x^t, y^t) = C(x,y)^t$ for all $x,y \in [0,1]$ and $t > 0$ in this case, and it can be rewritten as

$$C(x,y) = \exp\left( (\ln x + \ln y) A \left( \frac{\ln x}{\ln x + \ln y} \right) \right) \tag{3}$$

using a function $A : [0,1] \to [\frac{1}{2}, 1]$ called the Pickands dependence function (Pickands, 1981). The function $A$ is convex and satisfies $\max(w, 1-w) \leq A(w) \leq 1$. The extremal index is defined as two times its value at the point 0.5:

$$\theta_{x,y} := 2 \cdot A(0.5). \tag{4}$$

In the case of a perfect positive correlation between $X$ and $Y$ ($\mathrm{Corr}(X,Y) = 1$), the extremal index takes its minimal possible value of 1, for independent variables it reaches the maximal value of 2.

The extremal coefficient gives rise to a dissimilarity function

$$D_0(x,y) := \theta_{x,y} - 1 \tag{5}$$

To estimate it, we use the madogram estimator as described in Ribatet et al. (2015) and Cooley et al. (2006) and rewrite the extremal coefficient as

$$\theta_{x,y} = \frac{1 + 2\nu_{x,y}}{1 - 2\nu_{x,y}} \tag{6}$$

with the madogram $\nu_{x,y} = \frac{1}{2}\mathbb{E}[|F_X(X) - F_Y(Y)|]$. The madogram can be estimated by replacing $F_X, F_Y$ with their empirical counterparts. For a data sample $(x_1, y_1), \ldots, (x_n, y_n)$, we then obtain

$$\hat{\nu}_{x,y} = \frac{1}{2n(n+1)} \sum_{i=1}^{n} \left| \sum_{j=1}^{n} (\mathbf{1}_{x_j \leq x_i} - \mathbf{1}_{y_j \leq y_i}) \right|. \tag{7}$$



Note that the extremal coefficient is invariant under rank transformations and especially that it does not depend on the values

of the GEV parameters of the marginal distributions (in fact, in Ribatet et al. (2015) and Cooley et al. (2006) it was only used

in the special case of $\mathrm{GEV}(1,1,1)$ distributed margins, but it can easily be extended to the general case). It may be desirable

also to include the similarity of the estimated GEV parameters in the dissimilarity measure used for the clustering. As a further

generalised dissimilarity measure we propose

$$D_\lambda(x,y) := (1-\lambda)D_0(x,y) + \lambda\frac{1}{3}\Big(d_\mu(x,y) + d_\sigma(x,y) + d_\gamma(x,y)\Big) \tag{8}$$

with $d_p(x,y) = \frac{|p_x - p_y|}{\max_{a,b}|p_a - p_b|}$ the normalised distance between the parameter estimates at the points $x$ and $y$, where $p$ is one

of the parameters $\mu, \sigma, \gamma$ and with $\lambda \in [0,1)$ a weighting parameter.

Hierarchical clustering algorithms are well-known, for an introduction see Murtagh and Contreras (2012). To choose an

optimal number of clusters, we consider an approach by Salvador and Chan (2004) called the L-Method. In each step of the

hierarchical clustering, the two clusters with minimal dissimilarity are combined, therefore we can plot the number of clusters

versus the dissimilarity between them, resulting in a graph called the evaluation graph. The dissimilarity between clusters

necessarily grows as the total number of clusters is reduced. The idea of Salvador and Chan (2004) is to find a point from

which on the growth rate of the dissimilarity measure increases considerably. It can then be expected that the clusters up to

this point combine rather similar data points, while combining them to larger ones would yield artificial results. To determine

such a point of change, in the first step, a suitable range of the number of clusters is selected. For our example, we use ranges

starting with 10 and having no more than 550 clusters. Now, for each possible point of change $c$ in this range, the x-axis of the

graph is divided into the two parts to the left and the right of the change point, and a linear regression line is fitted to each of

the two partial graphs. The root mean squared errors (RMSE's) of the two regression lines are weighted with the number of

points involved in the regression analysis and summed up. The change point with the minimal combined RMSE is chosen as

the optimal cluster number. As an alternative method, we set the number of clusters to the highest possible number such that a

fixed threshold dissimilarity between clusters is not exceeded (Threshold method). This number can easily be read off on the

evaluation graph.

## 4  Results

We start with calculating for each annually maximised data point its empirical mean and standard deviation, as can be seen

in Fig. 1. In most regions, similar mean values can be observed. A notable overestimation of the annual maxima of monthly

precipitation by the climate model takes place in the Himalayas and along the western continent coasts of the Americas.

Underestimation occurs most prominently in the Amazon region and parts of Central America, as well as in Bangladesh and

East Asia. Looking at the standard deviation, a similar pattern as for the empirical mean can be observed, but with a stronger

tendency for underestimation, which occurs also in India and the northern part of Australia. In Fig. 2 a) and b), quantile-quantile

plots (QQ-plots) of empirical mean and standard deviation can be seen. The quantiles of the empirical mean are in general

**Figure 1.** The empirical mean (a, c, e) and empirical standard deviation (b, d, f) of the annual maxima of monthly precipitation of the CRU TS4.04 reanalysis data set (a, b) and of the AWI-ESM model data set (c, d) and their difference (reanalysis minus model data; e, f). Values exceeding the scale limits are truncated. Units are $\mathrm{mm/month}$.

similar, but the highest quantiles show a strong discrepancy. Regarding the standard deviation, this tendency is much more pronounced, corresponding to the larger areas of underestimation of empirical standard deviation we saw in 1. In Fig. 2 c), the difference in empirical mean and the difference in empirical standard deviation are plotted against each other, and it is clearly visible that overestimation (underestimation) of the empirical mean corresponds also to overestimation (underestimation) of the empirical standard deviation. A similar case of heteroscedasticity has also been noted in Lohmann (2018) when investigating Holocene climate.





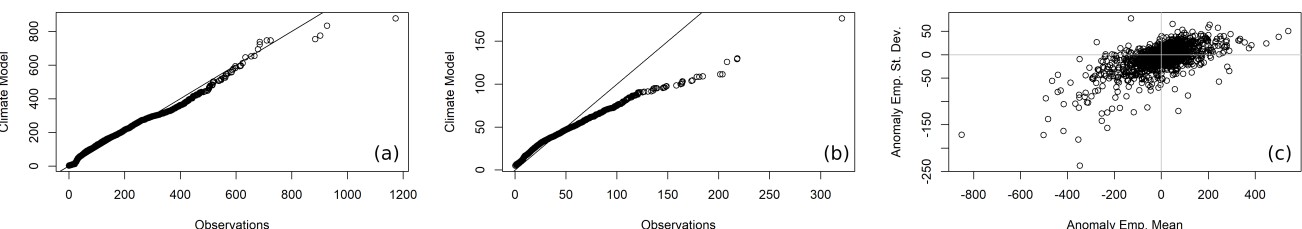

**Figure 2.** QQ-Plots comparing the empirical mean values (a) and the empirical standard deviations (b) and of the annually maximised monthly precipitation of the CRU TS4.04 reanalysis data set and of the AWI-ESM model data set. Deviance of empirical mean and standard deviation plotted against each other (c). Units are $\mathrm{mm/month}$.

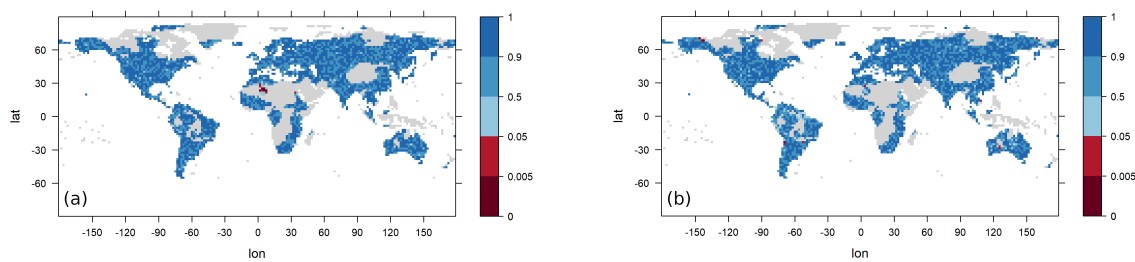

**Figure 3.** P-values of Kolmogorov-Smirnov tests for the hypothesis that the data follow a GEV distribution with parameters estimated using probability-weighted moments. Test results for the AWI-ESM climate model (a) and for the CRU TS4.04 reanalysis data (b).

As pointed out by Katz and Brown (1992), the frequency of extreme events is strongly influenced by changes (or, in this case, overestimation) of the mean as well as of the variance of a distribution. Therefore, a systematic over- and underestimation of extremes can be expected in certain regions based on the results in Figs. 1 and 2.

Fitting the GEV distributions to the data and applying KS-Tests to check the goodness of fit, the hypothesis of a GEV distribution with the estimated parameters is not rejected for nearly all grid points in both observational and climate model data, except for parts of the Sahara and some isolated points.

The three GEV parameters estimated are location, scale and shape, with location and scale very roughly corresponding to mean and variance, and the shape parameter yielding information about the degree of heavy-tailedness. The estimated parameter values are shown in Fig. 4. In Fig. 5 the differences between model and observations parameters are shown. Shaded areas are areas in which the model parameter falls into the $95\%$ confidence interval of the corresponding observation parameter and vice versa. We can observe a strong similarity between the anomaly of the location parameters and the anomaly of the empirical means discussed above, and likewise a similarity between the anomalies of scale parameters and empirical standard deviations. For the location parameter, we often observe high differences, and the parameters estimated for one data set seldom fall into the confidence interval derived from the other data set. The confidence intervals of the estimated scale parameters





are met more often, although there are also large regions with a high difference in the two estimates. The estimated shape parameters often lie within the confidence intervals, but it needs to be noted that the estimator of the shape parameter is known

to be sensitive to small variations in the data. Therefore, the confidence intervals calculated using the parametric bootstrap tend to be large and not particularly informative. In Fig. 6, the anomalies of the $95\%$ upper quantiles of the estimated GEV distributions are depicted, again with shading indicating areas lying in confidence levels determined using parametric bootstrap. Climate extremes are most strongly overestimated by the model in the mountanious the Himalaya, the Andes and the Rocky Mountains. An underestimation of climate extremes takes place most notably in the Amazon region and parts of eastern Asia.

This corresponds well to the regions of over- and underestimation of the empirical means and standard deviations and the implications of such misestimations discussed above.

We apply the hierarchical clustering algorithms using the two dissimilarity measures $D_0$ and $D_{0.25}$ as introduced in the previous section. The numbers of clusters determined using the L-Method with selected cluster ranges (from 10 to a maximal

number of clusters $m$) and using the threshold method with selected threshold dissimilarities $h$ is documented in Table 1.

**Table 1.** The number of clusters determined with the L-Method (above the middle line) and the threshold method (below the middle line) for different ranges/thresholds and for dissimilarity measure $D_0$ (left) and $D_{0.25}$ (right).

| $\mathbf{D_0}$ | Model | Obs. | $\mathbf{D_{0.25}}$ | Model | Obs. |
|---|---|---|---|---|---|
| $m = 250$ | 64 | 146 | $m = 250$ | 187 | 102 |
| $m = 300$ | 148 | 148 | $m = 300$ | 165 | 142 |
| $m = 400$ | 200 | 296 | $m = 400$ | 223 | 140 |
| $m = 500$ | 234 | 291 | $m = 500$ | 232 | 265 |
| $h = 0.85$ | 143 | 127 | $h = 0.675$ | 118 | 109 |
| $h = 0.825$ | 188 | 177 | $h = 0.65$ | 165 | 167 |
| $h = 0.8$ | 232 | 221 | $h = 0.625$ | 219 | 220 |
| $h = 0.775$ | 280 | 254 | $h = 0.6$ | 281 | 265 |

The results of the L-Method strongly depend on the value of $m$ and show a fairly inconsistent behaviour, making this method not very suitable for the comparison of two data sets. The threshold method consistently predicts a similar, but in most cases slightly lower cluster number for observational data than for climate model data. In Fig. 4, the clusters for both data sets are

depicted using the threshold method for dissimilarity measure $D_0$ with threshold $h = 0.825$ as well as for dissimilarity measure $D_{0.25}$ with threshold $h = 0.65$.



**Figure 4.** The estimated GEV parameters location (a, b), scale (c, d) and shape (e, f) for climate model data (a, c, e) and for reanalysis data (b, d, f). Values exceeding the scale limits are truncated. Units are $\mathrm{mm/month}$.



**Figure 5.** Difference between model and observational GEV parameter estimates: Location parameter (a), scale parameter (b) and shape parameter (c). Values exceeding the scale limits are truncated. Units are $\mathrm{mm/month}$.

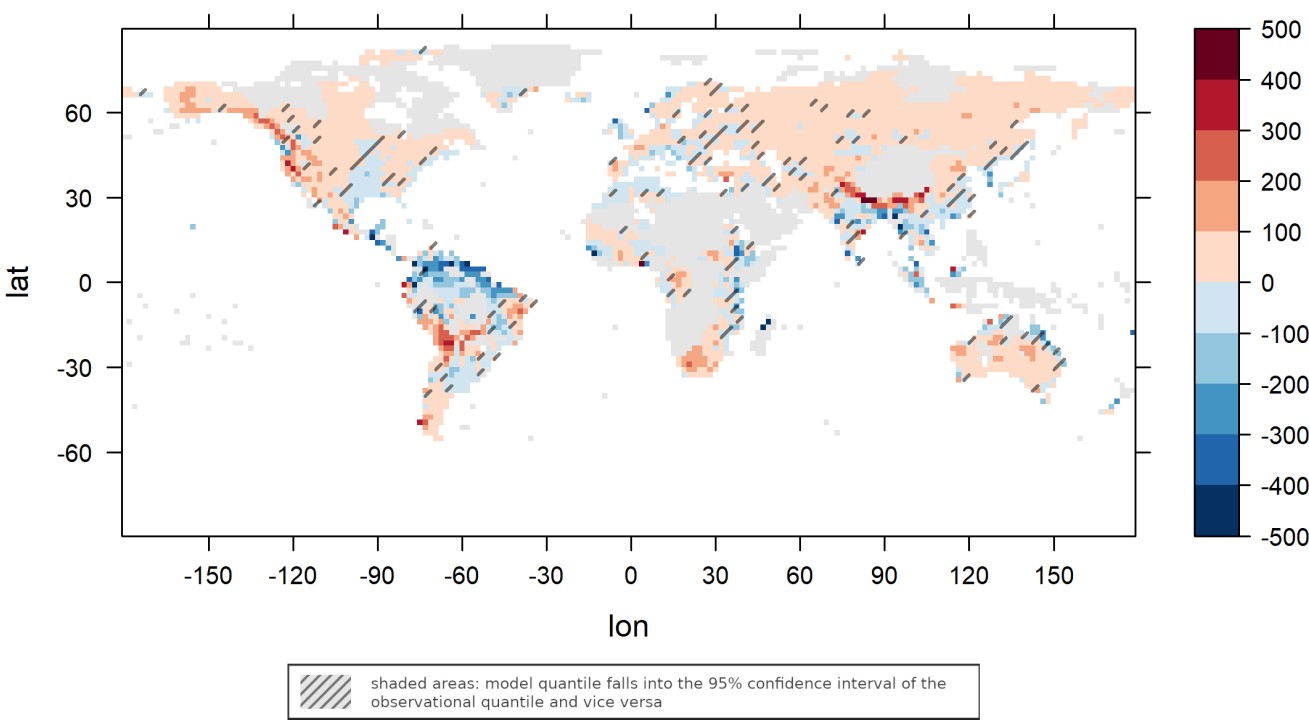

**Figure 6.** Difference of the 0.95-quantiles of the estimated GEV distribution for model and observational data. Values exceeding the scale limits are truncated. Units are $\mathrm{mm/month}$.

## 5   Conclusions

We presented approaches and methods to validate climate model outputs by comparing their extremal behaviour to the extremal behaviour of observational data. To illustrate these methods, we compared precipitation extremes between the AWI
Earth System Model and the CRU TS4.04 data set of reanalysed observations. After an analysis of empirical statistics, we fitted the data to GEV distributions and analyse the differences in estimated parameters. Then we continued with an analysis of spatial concurrence of extremes based on a hierarchical clustering approach and a dissimilarity measure derived from bivariate copula theory. While the empirical statistics are similar for many parts of the world, we can also identify larger regions of a continuous over- and underestimation of empirical means and standard deviations by the climate model. These misestimations
often go hand in hand with a similar misestimation of the standard deviation (heteroscedasticity), although for the standard deviation a stronger tendency for underestimation can be observed. Misestimations of mean and standard deviations translate into a misestimation of extreme values, and this can be confirmed by the comparison of the fitted GEV distribution parameters and the 0.95-quantiles derived from them. The shape parameter, indicative of the heavy-tailedness of the distribution, is in general similar between model and observational data, but because of the difficulties in reliably estimating this parameter from

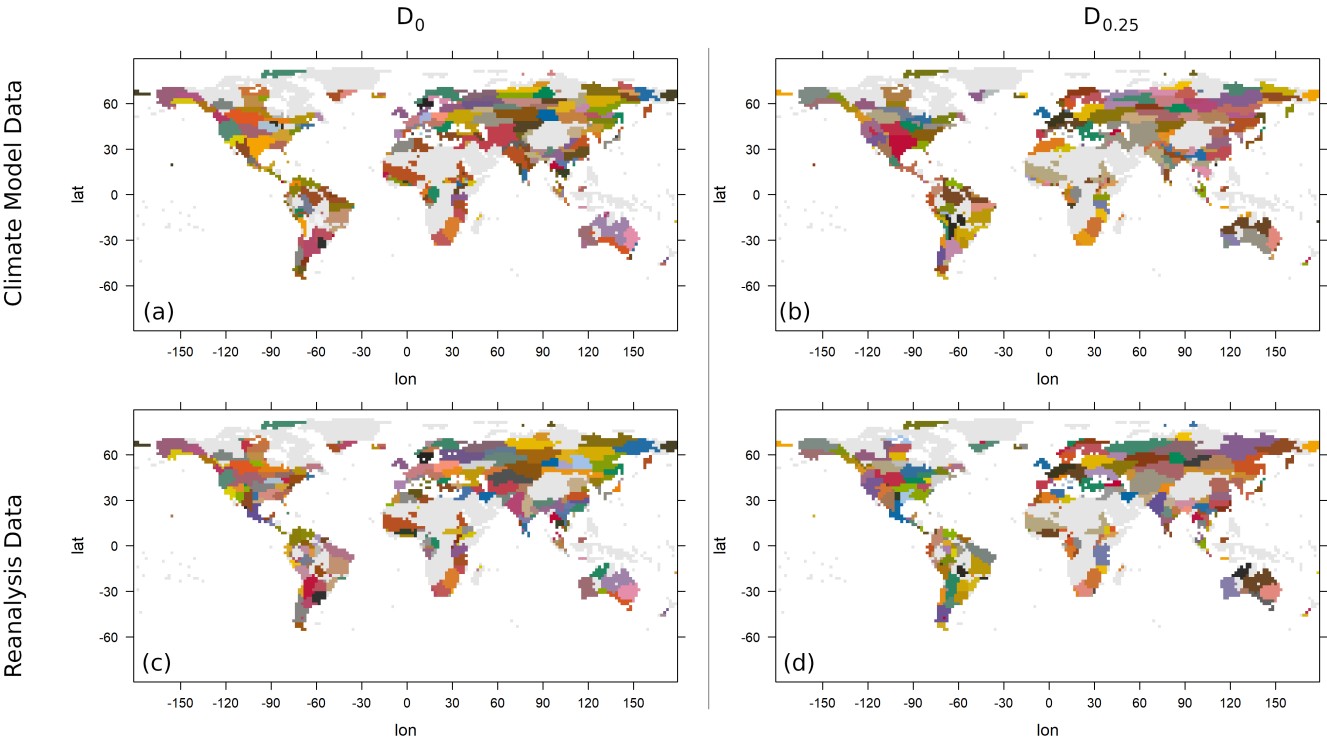

**Figure 7.** Clustering of model data (a, b) and observational data (c, d) with the dissimilarity measure $D_0$ and threshold $h = 0.825$ (a, c) and with dissimilarity measure $D_{0.25}$ and threshold $h = 0.65$ (b, d).

data (that are in turn a result of the rareness of extreme events in the data) these results have to be taken with caution.

The cluster analysis based on spatial dependencies and the occurence of concurrent extremes shows that there is generally a good agreement between identified clusters. Also the number of clusters is in general similar, with a slight tendency for a higher cluster number in the model data. Since it is mostly large-scale weather events and teleconnections contributing to concurrent climate extremes, this indicates that the basic physical behaviour underlying them is in general well captured by the AWI Earth

System Model. Further analysis can be conducted to investigate in detail the reasons for different clusterings over selected regions.

In this work, we use a clustering algorithm based on bivariate copulae to perform a spatial analysis of extreme values. Another possible approach is the direct application of multivariate copulae. While parametric copulae families are applicable

only to a very limited extent in high dimensions, the use of a non-parametric estimator based on Bernstein polynomials is a promising idea. This technique enabled Marcon et al. (2014) to estimate the common distribution of up to seven variables in their analysis of French precipitation data. Copulae based on Bernstein polynomials are also used in multivariate extreme





value analysis with a focus on multiple testing (Neumann et al., 2019). In global climate models, the number of dimensions is much higher than seven and the approach by Marcon et al. (2014) is not directly transferable. Based on another approach, the multivariate spatial distribution of precipitation has also been described using max-stable processes, first by Smith (1990) and Schlather (2002) and then extended by Opitz (2013) and Ribatet et al. (2015). This procedure is successfully used to model precipitation over Switzerland (Ribatet, 2017). The models based on max-stable processes assume spatial stationary (i.e. the spatial dependence between two points depends only on their distance). This assumption is justifiable for small regions like Switzerland, but it makes the models in their present form unsuitable for global data.

The clustering approach presented here focuses on the comparison of extremal events at different locations, thereby supplementing the analyses of climate extremes that are often focused on extremes at a specific location (Zhang et al., 2011). An application to daily data that has been annually or seasonally maximised, is also possible, but beyond the scope of this paper. Besides, the method can be used for the comparison of ensembles of models (see Sillmann et al., 2013; Kim et al., 2020). In addition to model validation, the definition of regions with concurrent extremes may turn out useful for assessments of risks in a economical context and for insurance. It needs to be noted, though, that extremes in climate models and in gridded reanalysis data sets tend to be damped because of the spatial averaging performed during the creation of the data (Bador et al., 2020). Another possible field of application is palaeoclimatology. Spatial distribution of precipitation extremes is known to have changed markedly in the past (Lohmann et al., 2020; Ionita et al., 2021b), and clustering based on climate models could be used to generalise the sparse existing data to larger regions.

*Code and data availability.* The CRU TS4.04 reanalysis data are available at https://catalogue.ceda.ac.uk/uuid/89e1e34ec3554dc98594a5732622bce9. The AWI-ESM climate model data used in this work is available under https://www.doi.org/10.22033/ESGF/CMIP6.9328. The software code (in R) used for the analyses can be found in the supplementary material to this paper.

*Author contributions.* Initial concept by TD and GL. JC led the writing of the paper and implemented the statistical data diagnostics. TD contributed to statistical methodology. GL contributed to the climatological analysis. All authors read and approved the manuscript.

*Competing interests.* The authors declare that they have no conflict of interest.

*Acknowledgements.* The authors are grateful to Manfred Mudelsee for constructive discussions and helpful suggestions. JC is funded through the Helmholtz School for Marine Data Science (MarDATA), Grant No. HIDSS-0005. GL receives funding through "Ocean and Cryosphere under climate change" in the Program "Changing Earth - Sustaining our Future" of the Helmholtz Society and PalMod through BMBF.



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
