# Peer review of "Variability and extremes: Statistical validation of the AWI-ESM"

_Geoscientific Model Development, 2021_

## Author Response (AR1)

Response to Anna Kiriliouk:

Thank you very much for your detailed and constructive review of our manuscript. In the following response, our answers to your comments are written in blue, and corrected or added parts of the manuscript are written in green.

The introduction does not seem to cite other papers that study model validation focused on extremes, yet I think it is already an active research topic. For an example, see for instance

*Timmermans, B., Wehner, M., Cooley, D., O'Brien, T., & Krishnan, H. (2019). An evaluation of the consistency of extremes in gridded precipitation data sets. Climate dynamics, 52(11), 6651-6670.*

The following additional paragraph citing relevant literature (Page 3, lines 66-74) was added:

Model validation in terms of precipitation extremes is already an active research topic. Tabari et al. (2016) investigate the performance of global and regional climate models using the peaks-over-threshold approach. An evaluation of regional and global climate models using extreme precipitation indices is conducted by Bador et al. (2020a), revealing a tendency for stronger extremes in regional models. A similar result was obtained by Mahajan et al. (2015) by comparing climate model and observational precipitation data over the United States using GEV distributions. Timmermans et al. (2019) conduct pairwise comparisons of the precipitation extremes of numerous gridded observation-based datasets and find considerable differences between the datasets especially in mountainous regions. Precipitation extremes over India are investigated by Mishra et al. (2014) using GEV distributions and comparisons of indices with a focus on changes over time.

Page 2, lines 35-36: please mention that there are two popular approaches, block maxima and peaks-over-thresholds.

The following paragraph describing the peaks-over-thresholds method was included (Page 2, lines 36-41):

Two common approaches are used to overcome this issue: peaks-over-threshold and block-maxima. In the peaks-over-threshold approach, a fixed threshold is selected. The distribution of the data exceeding this threshold can then be approximated by a generalised Pareto distribution if some additional assumptions are fulfilled (see McNeil et al. (2015), Chapter 7.2 for more details). The peaks-over-threshold approach is frequently applied in climatology and hydrology (Acero et al., 2011; Fowler and Kilsby, 2003; Kiriliouk et al., 2019).

Page 4, lines 110-111: I would remove ``especially for the yearly maximum of daily average precipitation'' since the GEV has been used extensively for many types of data.

We removed the passage.

Page 5, lines 131-132: this sentence is rather unclear, could you maybe reformulate?

We reformulated the sentence, it now reads (Pages 6, lines 181-184):

Then, if (X, Y) is the weak limit of block-wise maxima of a sequence of i.i.d. two-dimensional variables when the block size goes to infinity (a similar condition as in Sect. 3.1, extended to two-dimensional random variables), it follows that X and Y are (jointly) GEV distributed. It follows as well that the copula must fulfil $C(u^t; v^t) = C^t(u, v)$ for all u, v \in [0, 1] and t > 0 (see McNeil et al. (2015), Theorem 7.44 and 7.45). Such a copula is called max-stable and it can be written as [...].

Page 5, lines 145-146: please note that the lower bound of the extremal index corresponds to perfect dependence (comonotonicity), which is more general than a Pearson correlation of 1 (a Pearson correlation of 1 implies comonotonicity but not vice versa).

We have corrected the passage. It now reads (Pages 7, lines 189-192):

The extremal coefficient takes its minimal possible value of 1 if X and Y are comonotonic (so in particular it holds $\theta_{X,X} = 1$ for all X). The maximal possible value of 2 is obtained if X and Y are independent.

Page 6, equation (8): does it make sense to « weight » the three marginal parameters equally? Maybe the shape parameter could play a bigger role than the mean and scale parameters?

This is an interesting idea. We have considered alternative weightings, but it was hard to assess whether they really gave better results. Furthermore, the shape parameter estimators have a much higher variability compared to location and scale estimators, therefore we think putting to much weight on it would not be helpful. We therefore decided to stay with the weights we used initially, and added the following additional paragraph (Page 7-8, lines 212-218) describing the issue:

Instead of an equal weighting, it would also be possible to use different weights for $d_\mu$, $d_\sigma$ and $d_\gamma$, but the selection of a set of weights that is clearly better suited to describing GEV distribution dissimilarity is difficult. It could be argued to put more weight on the shape parameter since this parameter describes the heavy-tailedness of the distribution and therefore the strength of its extremes relative to the non-extreme values. On the other hand, we will see in the next section that the uncertainty in the shape parameter estimation is considerably higher than the uncertainty in the estimation of the other two parameters at least for our data, which would speak against weighting shape parameter differences too strongly.

Page 9, line 210: it is not necessary to use bootstrap-based confidence intervals for the PWM estimators, since their asymptotic covariance is known and has a simple expression; see

*Hosking, J. R. and J. R. Wallis (1987). Parameter and quantile estimation for the generalized Pareto distribution. Technometrics 29(3), 339–349.*

*Ribereau, P., P. Naveau, and A. Guillou (2011). A note of caution when interpreting parameters of the distribution of excesses. Advances in Water Resources 34(10), 1215– 1221.*

Thank you very much for the suggestion. We agree that the PWM estimators are advantagous especially with regard to computation time. When applying them, we noted however that for several time series, large differences between bootstrap and PWM confidence intervals arose, especially when estimating confidence intervals for the shape parameter (see Fig. 1 below).

[Figure]

[Figure]

Fig. 1: Comparison of the ranges of the GEV parameter confidence intervals using PWME normal approximation and using bootstrap

Indeed, if the estimated shape parameter takes certain values (too far away from zero), the PWM CIs are known to have a very high bias and variance, see Hosking, Wallis and Wood (1985). Also in our data, the strong discrepancies between the two types of confidence intervals appeared for certain values of the shape parameter (see Fig. 2).

[Figure]

Fig. 2: Difference of the CI ranges plotted against the estimated value of the shape parameter

Since the conditions for applying PWM estimators are not fulfilled for all time series and because we calculate bootstrapped samples anyway for the calculation of CIs for the 95% quantiles, we

decided to stay with the bootstrap approach for all time series. We added the following paragraph about the issue in the paper (Page 6, lines 157-166):

We also use the parametric bootstrap method with 2500 resamples to compute 95% confidence intervals for each GEV parameter and for the 95% quantiles of the distributions. Confidence intervals for the GEV parameters based on asymptotic normality also exist for the probability-weighted moments estimators, but, as shown by Hosking et al. (1985), they have a high bias and variance if the shape parameter is far away from zero. In our data, for several time series such a value is estimated for the shape parameter, and comparisons between the confidence intervals based on bootstrap and those based on asymptotic normality also confirmed large differences in these cases. For the sake of methodological consistency and because we also use the bootstrap for the confidence intervals of the 95% quantiles, we calculated the GEV parameter confidence intervals using bootstrap for all time series. Since this method is quite time-consuming, it could also be advocated to choose the method of confidence interval calculation based on the estimated shape parameter value.

Page 13, Fig 7: the graphs are difficult to compare because of the many clusters and colors. Could you please resume the main differences in the text?

The following paragraph describing the images was added (Page 15, lines 285-295):

To exemplify the differences and similarities in the clusterings, we have a closer look at Europe in the $D_0$-clusterings. In the model data, there is one cluster covering western Spain and Portugal, one cluster covering eastern Spain, and one cluster consisting of southern France and Italy. Great Britain and Denmark are in the same cluster, the northern parts of France together with Belgium and the Netherlands in another one. One cluster covers Germany and Switzerland, and in Eastern Europe we see several clusters covering larger areas in the longitudinal direction, for example one cluster over Poland, one over Ukraine, and one over Turkey and Greece. The clusters in the observational dataset show a slightly different picture: Here, the whole Iberian Peninsula is in one cluster, and one large cluster extends over northern France, Belgium, the Netherlands and Germany to the western parts of Poland. On the other hand, Great Britain and Denmark are now in two separate clusters. Regarding other parts of the world, it is worth noting that in all four clusterings a large cluster cluster covering the Sahara (or at least all parts of it for which there are observations available) can be identified. There are no clusters extending over two regions that are very far apart from each other, and in general clusters tend to cover more area in the longitudinal direction than in the latitudinal one.

Page 13, line 250: ``While parametric copula families are applicable only to a very limited extent in high dimensions…'' I disagree, there are many possibilities to model high-dimensional data using parametric copulas, for example, through vine copulas.

You are right, thank you for the correction. We removed the sentence and referenced some applications of parametric copulas to precipitation data instead (Page 20, lines 350-353). In order to be able to refer to max-stable models and spatial spationarity in this paragraph, the description of max-stable models that originally followed later in the chapter was moved up.

  Castro-Camilo and Huser (2020) created a model for the spatial distributions of extreme tail dependencies based on factor copulae, allowing them to use the relaxed assumption of local spatial stationarity and therefore to apply their model to the whole contiguous United States. From the area

of parametric copulae, also vine copulae have been employed to model precipitation data by Vernieuwe et al. (2015) and by Nazeri Tahroudi et al. (2021).

Page 14, line 259: aren't the spatially stationary method well suited to model the clusters as identified previously?

This is a very good idea. It would be beyond the scope of this paper to test the application of those models to the data thoroughly, but we mentioned it in the paper as an option for further research (Page 21, lines 363-366):

In order to investigate extreme precipitation within the area covered by one cluster in more detail, the spatially stationary max-stable models or the copulae-based models mentioned above could be employed. Most of the clusters cover only a small region, therefore spatial stationarity might be a reasonable assumption, although it is not a direct consequence of the data being in the same cluster.

Typographical errors:

- Page 4, line 114: resultung -> resulting

- Page 9, line 224: Fig 4 should be Fig 7?

Thank you for pointing them out, we corrected the errors.

Response to Qingxiang Li:

Thank you very much for your review of our manuscript and for the corrections and suggestions. We have answered to your comments in blue, and added parts of the manuscript are written in green.

1) The caption of Fig 1 does not agree with the figs.

You are right, the figure itself was correct but the caption was wrong. We corrected the error.

2) Since only one model are used in this manuscript, I would suggest the authors choose a model that agree most to the observations. What you need to do is to compare the performance among all available models first, and then apply the methods to that model output. This would more interesting.

We followed your suggestion and designed a measure (called Average Weighted Quantile Difference; AWQD) to compare the performance of different climate models. The formula we used for this measure is described on Page 6, lines 168-173. We evaluated a set of different CMIP6 models and compared their performance. It turned out that in general, models with a higher spatial resolution tend to perform better. The CMIP6 model with the best performance was found to be the model EC-Earth3-Veg-LR, and in addition to the AWI-ESM, we presented our results also for it (Page 15-16, lines 297-318):

For the AWI-ESM, we calculated an AWQD of 52.98, making it the third-best of all 27 CMIP6 models analysed. A full table of the models and their AWQDs is provided in the supplement to this paper. In Fig. 8, the AWQDs are plotted against the model resolution (the total number of model grid points in units of 104). A linear regression (red line; intercept: 73.310, slope: -2.368) indicates that models with a higher resolution have a tendency to describe extremal precipitation better. A test on the significance of the slope parameter (null hypothesis of the slope parameter being equal to zero) was significant at the 5% level with a p-value of 0.0357. The best model in terms of the AWQD is the high-resolution model EC-Earth3-Veg-LR (EC-Earth Consortium, 2020) with a value of 44.71. We will now discuss results for this model in more detail, while results for the other models can be found in the supplement. For the EC-Earth3-Veg-LR, the estimated GEV parameters and anomalies are shown in Fig. 9. The differences of the 95% quantiles are depicted in Fig. 10. The numbers of clusters determined using the L-Method and the threshold method are found in Table 2 and images of clusterings are shown in Fig. 11. [...]

Furthermore, we have added new Figures (Fig. 8 through 11), extended the introduction (Page 3, lines 61-64) and made several smaller text adaptations to the manuscript because of the additional content. We also added a supplement to the paper, in which figures corresponding to the analysis of the EC-Earth3-Veg-LR model (that were omitted in the main text because they were similar to figures already shown) are presented (Fig. S1 through S3). A table listing the results of the model comparison was also included in the supplement, as well as figures (Fig. S4 onwards) showing the main results for each of the other considered CMIP6 models.

Further changes to the manuscript:

In addition to the corrections mentioned above and some minor text corrections, we had to correct an error in the description of our analysis procedure (Page 4, line 119-120; reanalysis data are interpolated to the climate model grid, not the other way around). Furthermore, we reworked the notations in Section 3.2 because sometimes the same notations/indices were used for different objects. In the same section, we also rearranged a few paragraphs in order to have a clearer separation of the copula theory and its application in the clustering. We feel that the section is now easier to read.